# Myrtinols A–F: New Anti-Inflammatory Peltogynoid Flavonoid Derivatives from the Leaves of Australian Indigenous Plant *Backhousia myrtifolia*

**DOI:** 10.3390/molecules28052160

**Published:** 2023-02-25

**Authors:** Shintu Mathew, Kenneth Zhang, Xian Zhou, Gerald Münch, Francis Bodkin, Feng Li, Ritesh Raju

**Affiliations:** 1NICM Health Research Institute, Western Sydney University, Penrith, NSW 2750, Australia; 2Department of Pharmacology, Western Sydney University, Campbelltown Campus, Sydney, NSW 2560, Australia; 3School of Science, Western Sydney University, Penrith, NSW 2751, Australia

**Keywords:** *Backhousia myrtifolia*, aboriginal knowledge, peltogynoid derivative, nitric oxide, flavonoids, plants

## Abstract

Our in-house ethnopharmacological knowledge directed our anti-inflammatory investigation into the leaves of *Backhousia mytifolia*. Bioassay guided isolation of the Australian indigenous plant *Backhousia myrtifolia* led to the isolation of six new rare peltogynoid derivatives named myrtinols A–F (**1**–**6**) along with three known compounds 4-*O*-methylcedrusin (**7**), 7-*O*-methylcedrusin (**8**) and 8-demethylsideroxylin (**9**). The chemical structures of all the compounds were elucidated by detailed spectroscopic data analysis, and absolute configuration was established using X-ray crystallography analysis. All compounds were evaluated for their anti-inflammatory activity by assessing the inhibition of nitric oxide (NO) production and tumor necrosis factor- α (TNF-α) in lipopolysaccharide (LPS) and interferon (IFN)-γ activated RAW 264.7 macrophages. A structure activity relationship was also established between compounds (**1**–**6**), noting promising anti-inflammatory potential by compounds **5** and **9** with an IC_50_ value of 8.51 ± 0.47 and 8.30 ± 0.96 µg/mL for NO inhibition and 17.21 ± 0.22 and 46.79 ± 5.87 µg/mL for TNF-α inhibition, respectively.

## 1. Introduction

Inflammation is the body’s natural defense response to various harmful stimuli including pathogens, heat, toxic chemicals, and injuries [1]. During the initial stage of trauma or infection, body initiates various cellular and molecular events which include the secretion of many proinflammatory cytokines and chemokines such as interleukin-6 (IL-6), interleukin-1β (IL-1β), tumor necrosis factor-α (TNF-α) as well as reactive oxygen species (ROS) and nitric oxide (NO) to restore tissue homeostasis and resolve acute inflammation [2,3]. The pro-inflammatory cytokines and/or bacterial lipopolysaccharides (LPS) can activate inducible nitric oxide synthase (iNOS) to produce continuously high concentrations of NO, which can then further induce injury at inflammatory sites [4]. Therefore, suppressing the overproduction of NO appears to be a promising strategy for the control of inflammatory diseases.

*Backhousia myrtifolia* (Myrtaceae) which is commonly referred to as carrol, neverbreak, iron wood, grey myrtle or cinnamon myrtle is a small rainforest tree species that grows in subtropical rainforests regions of Eastern Australia [5,6,7]. It was first discovered and subsequently used by the Indigenous communities of Australia, where the leaves were used to treat colic babies. *B. myrtifolia* is also known to harbor oils that have cinnamon-like aroma displaying both anti-bacterial and anti-fungal properties [8].

Our ongoing search to discover new anti-inflammatory molecules led us to explore the leaves of *B. myrtifolia* resulting in the isolation and identification of six new peltogynoid type flavonoids, myrtinols A–F (**1**–**6**), along with three known compounds, 4-O-methylcedrusin (**7**), 7-O-methylcedrusin (**8**), and 8-demethylsideroxylin (**9**) (Figure 1).

## 2. Structural Elucidation

Myrtinol A (**1**) was obtained as a green sticky mass. HRESI (−) MS analysis displayed a [M − H]^−^ deprotonated molecular ion peak at *m*/*z* 357.0959, corresponding to a molecular formula of C_19_H_17_O_7_ requiring 11 double bond equivalents (DBEs). NMR data (600 MHz, CD_3_OD) (Table 1, Figure 2) revealed the presence of one olefinic methyl (δ_H_ 2.05), one methoxy (δ_H_ 3.71), three distinct methylenes, including a methylene dioxy, H_2_-8’ (δ_H_ 5.91, δ_C_ 101.9) and an oxy-methylene, H_2_-7’ (δ_H_ 4.89, 4.73, δ_C_ 65.8) and a more traditional downfield methylene H_2_-4 (δ_H_ 3.11, 2.63, δ_C_ 28.1). The presence of a downfield oxy-methine H-2 (δ_H_ 4.53, δ_C_ 74.6) and a traditional oxy methine H-3 (δ_H_ 3.64, δ_C_ 73.1) was also evident. The presence of two aromatic resonances (δ_H_ 6.21) and (δ_H_ 6.66) both as singlets showed several HMBC correlations to different quaternary carbons and oxy-carbons suggesting the presence of two separate penta-substituted benzene systems. The aromatic proton H-8 (δ_H_ 6.21) from the first of the benzene system (ring A) showed HMBC correlations to C-6 (δ_C_ 110.5), C-7 (δ_C_ 155.2), C-9 (δ_C_ 152.9) and C-10 (δ_C_ 105.1). The olefinic methyl (δ_H_ 2.05, δ_C_ 8.6) was assigned to be attached at C-6 in ring A based on having HMBC correlations to C-5 (157.6), C-6 and C-7 (Figure 1). The only methoxy (δ_H_ 3.71) in the structure was also assigned to ring A based on a HMBC correlation to C-5 (Figure 1). The aromatic proton H-2’ (δ_H_ 6.66) from the second benzene system (ring B) showed HMBC correlations to C-1’ (δ_C_ 117.7), C-3’ (δ_C_ 133.4), C-4’ (δ_C_ 147.6) and C-2 (δ_C_ 74.6). These correlations indicated the existence of another heavily substituted benzene ring. HMBC correlations from the methylene dioxy protons H_2_-8’ (δ_H_ 5.91, δ_C_ 101.9) to C-3’ and C-4’ confirmed the attachment of this ring system (ring E) onto the benzene ring generating a 1,3-benzodioxole moiety. COSY correlations revealed a single isolated spin system consisting of two oxymethines (H-2 (δ_H_ 4.53, δ_C_ 74.6), H-3 (δ_H_ 3.64, δ_C_ 73.1) and a downfield methylene H_2_-4 (δ_H_ 3.11, 2.63, δ_C_ 28.1). With the current structural fragments accounting for 9 out of the 11 DBEs, the situation required making room for two additional ring systems to be incorporated in the structure (Figure 1). Key diagnostic HMBC correlations from H_2_-4 to C-9 and C-10 and H_2_-7’ to C-1’, C-5’, C-6’ and C-3 confirmed the existence of two heterocyclic ring systems (rings C and D) where ring D was connected to ring B via an oxymethylene bridge representative of the peltogynoid type flavonoid system.

Myrtinol B (**2**) was obtained as a green solid. HRESI (+) MS analysis revealed a pseudomolecular ion [M + Na]^+^ ion peak at *m*/*z* 397.1247 ([M + Na]^+^, calcd 397.1263) in accordance with the molecular formula of C_20_H_22_O_7_Na requiring ten double bond equivalents. It was evident early on from the NMR spectrum and data that (**2**) was a close structural analogue of myrtinol A (**1**), and comparison of the spectroscopic data of **1** with **2** showed that **2** retained the ring system A–D. The only difference was the presence of two additional methoxy resonances at (δ_H_ 3.88 and δ_H_ 3.79) exhibiting HMBC correlations to C-3′ (δ_C_ 136.2) and C-4′ (δ_C_ 153.2), respectively, with the absence of the methylene dioxy resonance which supported the opening up of ring E (Table 1, Table 2 and Appendix A and Figure 3).

In addition to this, we were successful in generating a crystal for compound **2**, which was then subjected to X-ray crystallography and the data obtained assisted in confirmation of the structure (Figure 4, Appendix A) and helped resolve the absolute stereochemistry for the chiral centers C-2 and C-3 as *R* and *S.* From the NMR data alone, a large coupling of *J*_H-2/H-3_ (*9.5* Hz) also suggested a *trans* confirmation.

Myrtinol C (**3**) was obtained as a white solid. Its molecular formula was determined as C_20_ H_21_ O_8_ from HRESI (−) MS ion analysis at *m*/*z* (389.1246 ([M − H]^−^, calcd 389.1236). A close comparison of the spectroscopic data of **3** with **2** confirmed that myrtinol C possessed the same ring system as myrtinol B, with the only exceptions being the positional change of the hydroxy and methoxy groups present in the ring A and the introduction of a hydroxy functionality C-4 (δ_H_ 5.01, δ_C_ 70.6). This above structural changes were supported by HMBC correlation from H-4 (δ_H_ 5.01) to C-3 (δ_C_ 78.3), C- 9 (δ_C_ 154.0) and C-10 (δ_C_ 103.8), from H-8 (δ_H_ 6.15) to C- 6 (δ_C_ 106.8), C- 7 (δ_C_ 160.0), C-9 and C-10, from H-6 (δ_H_ 1.98) to C-5, C-6, C-7, and from H-7 (δ_H_ 3.77) to C-7 (Table 1, Table 2 and Appendix A and Figure 3). A large coupling of *J*_H-3/H-4_ (8.3 Hz) was suggestive of a *trans* confirmation, placing the absolute stereochemistry of C-4 to be *R*.

Myrtinol D (**4**) was obtained as a white solid. The molecular formula was revealed as C_20_H_21_O_7_ based on HRESI (−) MS ion analysis at *m*/*z* 373.1284 ([M − H]^−^, calcd 373.1287), which was identical to myrtinol B (**2**). Myrtinol D (**4**) was identified as a structural isomer of (**2**) with the only difference being the positional change between the meta coupled hydroxy and methoxy functionalities in ring A of the benzene unit. This was supported by HMBC correlations from 7-OMe (δ_H_ 3.76) to C-7 (δ_C_ 158.5) and H-8 (6.19) to C-7.

Myrtinol E (**5**) was obtained as a yellow solid. The molecular formula was revealed as C_20_H_19_ O_7_ from HRESI (−) MS ion analysis at *m*/*z* 371.1122 ([M − H]^−^, calcd 371.1131. Myrtinol E (**5**) showed the closest similarity to its analogue myrtinol A (**1**). The only difference in the NMR data and spectra was the detection of an additional methoxy resonance, which was confirmed to be attached at C-7 based on HMBC correlations of 7-OMe (δ_H_ 3.80) to C-7 (δ_C_ 158.8) and H-8 (δ_H_ 6.36) to C-7.

Myrtinol F (**6**) was obtained as a green sticky mass. The molecular formula was revealed as C_21_H_23_O_7_ from HRESI (−) MS ion analysis at *m*/*z* 387.1454 ([M − H]^−^, calcd 387.1444. Comparison of the spectroscopic data of **2** and **6** showed that myrtinol F possessed the same ring system as myrtinol B, with the only change being the replacement of hydroxy substituent at C-7 by a methoxy group, which was supported by HMBC correlation from H-8 (δH 6.36) C- 7(δ_C_ 157.8), and 7-OMe (δ_H_ 3.80) to C-7.

We also take into account that the level of purity of myrtinol F was not 100%, with the possibility of a terpene like impurity present in this fraction (Appendix A). Low yields and the level of difficulty experienced separating this two-compound mixture prevented us from obtaining an absolutely pure sample of myrtinol F. However, the HRMS data and the clear key NMR resonances attributed to myrtinol F allowed for its complete structural assignment.

The absolute stereochemistry for the chiral centers C-2 and C-3 for the remaining myrtinol analogues was assigned as *R* and *S,* respectively, based on the large coupling constant between *J*_H-2/H-3_ which was suggestive of a *trans* configuration along with the consideration of a likely biosynthetic relationship to myrtinol B (**2**) (Table 1 and Table 2) [9,10].

In addition to the discovery of the new myrtinols, three known compounds 4-*O*-methylcedrusin (**7**), 7-*O*-methylcedrusin (**8**) and 8-demethylsideroxylin (**9**) were identified by interpretation of their spectroscopic data (Appendix A) and a close comparison with published data [11,12,13,14].

## 3. Anti-Inflammatory Activity

All compounds were assessed for their anti-inflammatory activity by evaluating the inhibition of NO production and TNF-α in LPS plus interferon (IFN)-γ activated RAW 264.7 macrophages. All compounds were also evaluated for their cytotoxicity using the Alamar blue assay (Table 3).

## 4. Discussion

As shown in Table 3, the varying anti-inflammatory activity depended mostly on the functional groups attached to rings A, B and C. Based on the slight structural variations around the tetracyclic backbone of myrtinols A–F (**1**–**6**), we have attempted to evaluate the observed structural-activity relationship (SAR) among them. A SAR was observed between compounds **1** and **5** (both having a 1,3-benzodioxole moiety attached to ring B) which showed promising inhibition of NO production and TNF-α production, with a IC_50_ values of 11.47 ± 0.14, 24.54 ± 0.28 and 8.51 ± 0.47, 17.21 ± 0.22 µg/mL, respectively. In comparison, compounds **2** and **6** showed NO production inhibition with an IC_50_ value of 16.25 ± 0.77 and 12.62 ± 0.26 µg/mL, and TNF α production inhibition with an IC_50_ value 52.35 ± 7.47, and 30.55 ± 5.01 µg/mL, respectively, suggesting that the presence of 1,3-benzodioxole moiety (ring E) and methoxy group at C-5 and C-7 might play an important role in their anti-inflammatory activity. Interestingly, when comparing the SAR between compounds **3** and **4**, the only change between them being the introduction of a hydroxyl group at C-4 in **3**, this rendered the molecule inactive compared to the baseline activity observed for compound **4** where the NO inhibition was determined to be 29.31 ± 10.95 µg/mL.

We are mindful of the fact that the experimental NO inhibition values obtained for myrtinols may not be a true reflection of their anti-inflammatory activity profile mainly due to the fact that they mostly have a low therapeutic index associated with them. This suggests that future investigations on assessing their cytotoxic activity need to be performed in order to re-evaluate their potential as cytotoxic agents.

Among the known compounds, Compound 9 displayed interesting activity with an IC_50_ value 8.30 ± 0.96 µg/mL and IC_50_ value of 46.79 ± 5.87 µg/mL, whereas **7** and **8** did not show good anti-inflammatory activity (Table 3).

Peltogynoids are a rare type of cyclized flavonoid, where a carbon atom is attached between 3-OH and C-2′ of ring B [9,15]. Robinson and Robinson isolated the first peltogynoid type flavonoid, peltogynol in 1935 [16]. Phytochemical studies of Australian and South African *Acacia* species including *A. carnei* by Roux and co-workers identified additional peltogynoid derivatives, (+)-2,3-trans-3,4- trans- peltogynols, (+)-2,3-trans-3,4-cis-peltogynol, (+)-2,3-trans-peltogynone, carnein, peltogynin, β-photomethylquercitin [17]. Peltogynoid derivates have been reported to be isolated from different genera and species including *Goniorrhachis marginatu Taub* [18], *Pltogyne caringae*, *P. confertiflora*; *P. paniculate* [19], *Caesalpinia pulcherrima* [20], *Acacia nilotica* (L.) Delile [21], *Phytolacca icosandra* [22], *Bougainvillea spectabilis *[23]. The new peltogynoid derivatives reported in this paper bear the closest resemblance to the reported pubeschin analogues [24].

## 5. Experimental Section

### 5.1. General Experimental Procedures

UV spectra were recorded on an Agilent Carry UV-Vis Multicell Peltier spectrometer. NMR spectroscopic data were recorded on a Bruker Avance 600 MHz spectrometer (Bruker Biospin GmbH, Germany). HRMS (High Resolution Mass Spectrometry) was carried out using a Waters SYNAPT G2-Si mass spectrometer operating in the positive and negative ESI mode.

### 5.2. Plant Material

The leaves of *B. myrtifolia* were collected from the Australian Botanic Garden at Mount Annan (NSW, Australia). A voucher specimen (2005-0104) has been deposited at the Australian Botanic Gardens, at Mount Annan, NSW, Australia.

### 5.3. Extraction and Bioactivity-Directed Isolation

The fresh leaves of *B. myrtifolia* (300 g) were crushed using a hand blender and extracted sequentially using organic solvents based on their polarity (n-hexane, dichloromethane (DCM), ethyl acetate (EtOAc), ethanol (EtOH), methanol (MeOH), and finally, water) using a Buchi-811 Soxhlet Extraction system. Immediately after the initial stages of sequential fractionation, each corresponding fraction was subjected to anti-inflammatory screening using the inhibition of NO in LPS plus IFN-γ treated RAW 264.7 macrophages following the Griess test. The most active extract (DCM) (Appendix A) was resuspended in EtOH and was then later subjected to semi-preparative HPLC (Agilent 1260 Infinity II series) using an Agilent C_18_ column (5 µm, 250 × 9.4 mm) eluting at 1.8 mL/min from 10% MeCN/H_2_O to 100% MeCN (with a constant 0.01% FA (formic acid) modifier) over 60 mins and held for a further 6 mins and then equilibrated back to 10% MeCN/H_2_O in 1 min and maintained at 10% MeCN/H_2_O for an additional 3 mins, to give 17 fractions (Fr. 1–17) which included four pure fractions (Fr. 4 (**7**), Fr.8 (**1**), Fr.9 (**2**), and Fr.15 (**8**), corresponding to the following compounds: **7** (6.6 mg, *t*_R_ 24.7 min), **1** (5.4 mg, *t*_R_ 39.1 min), **2** (4.2 mg, *t*_R_ 41.7 min), and **8** (2.5 mg, *t*_R_ 50.3 min).

Fr.11 was re-purified by SB-C3 semipreparative column, (250 × 9.4 mm) using a gradient system of 50–60% MeCN/H_2_O with a flow rate of 2mL/min (with a constant 0.01% FA modifier) over 30 min to afford compounds **3** (1.4 mg, *t*_R_ 22.4 min), and **4** (2.0 mg, *t*_R_ 23.1 min). Fr.16 was purified using SB-Phenyl semipreparative column, (250 × 9.4 mm) with a gradient system of 10–80% MeCN/H_2_O over 15 mins followed by a change from 80–100% MeCN/H_2_O over 9 mins with a constant flow rate of 2 mL, and then held at 100% MeCN for 2 mins and equilibrated back to 10% MeCN/ H_2_O in 1 min and maintained at this gradient for an additional 2 min, to afford compound **5** (1.9 mg, *t*_R_ 21.8 min). Fr.17 was repurified using a semi-preparative Agilent C_18_ column (5 µm, 9.4 × 250 mm) by employing a gradient system of 10–90% MeCN/H_2_O over 15 mins followed by a gradient change of 90–100% MeCN/ H_2_O for an additional 5 min, and then held at 100% MeCN for an additional 2 min followed by an equilibration to 10% MeCN/H_2_O in 3 mins, affording compound **6** (6.4 mg, *t*_R_ 18.3 min). Compound **9** (4 mg, *t*_R_ 13.4 min) was acquired from Fr.5 using SB-Phenyl semipreparative column, (250 × 9.4 mm) eluting with 10–55% MeCN/H_2_O (with a constant 0.01% FA modifier) over 8 min at 2 mL/min, followed by a gradient change from 55–70% MeCN/H_2_O for an additional 7 min, and finally with a 70–100% MeCN/H_2_O (with a constant 0.01% FA modifier) over 2 min and then equilibrated back to 10% MeCN within 1 min and maintained 10% MeCN/H_2_O for an additional 2 min.

Myrtinol A (**1**): Greenish sticky mass [α]^25^_D_ +172.9 (c 0.001, MeOH); UV-Vis λ_max_ (MeOH) nm (log ε) 208 (5.31), 232 (5.37) and 280 (5.44); 1D and 2D NMR (600 MHz, CD_3_OD) data (see Table 1, Table 2 and Appendix A); HRESI (−) MS *m*/*z* 357.0959 [M − H]^−^ (calcd for C_19_H_17_O_7_^−^; 357.0974).

Myrtinol B (**2**): Green crystal (DCM:MeOH) [α]^25^_D_ +178.3 (c 0.001, MeOH); UV-Vis λ_max_ (MeOH) nm (log ε) 208 (5.31), 233 (5.37) and 281 (5.45); 1D and 2D NMR (600 MHz, CD_3_OD) data (see Table 1, Table 2 and Appendix A); HRESI (+) MS *m*/*z* 397.1247 [M + Na]^+^ (calcd for C_20_H_22_O_7_Na; 397.1263).

Myrtinol C (**3**): White solid [α]^25^_D_ +176.1 (c 0.0003, MeOH); UV-Vis λ_max_ (MeOH) nm (log ε) 208 (5.31), 233 (5.37) and 280 (5.44); 1D and 2D NMR (600 MHz, CD_3_OD) data (see Table 1, Table 2 and Appendix A); HRESI (−) MS *m*/*z* 389.1246 [M − H]^−^, (calcd for C_20_ H_21_ O_8_ 389.1236).

Myrtinol D (**4**): White solid [α]^25^_D_ +174.6 (c 0.0005, MeOH); UV-Vis λ_max_ (MeOH) nm (log ε) 208 (5.31), 232 (5.37) and 280 (5.44); 1D and 2D NMR (600 MHz, CD_3_OD) data (see Table 1, Table 2 and Appendix A); HRESI (−) MS *m*/*z* 373.1284 [M − H]^−^, (calcd for C_20_H_21_O_7_^−^; 373.1287).

Myrtinol E (**5**): Yellow solid [α]^25^_D_ +171.7 (c 0.0005, MeOH); UV-Vis λ_max_ (MeOH) nm (log ε) 208 (5.31), 232 (5.37) and 280 (5.44); 1D and 2D NMR (600 MHz, CD_3_OD) data (see Table 1, Table 2 and Appendix A); HRESI (−) MS *m*/*z* 371.1122 [M − H]^−^, (calcd for C_20_ H_19_ O_7_^−^; 371.1131).

Myrtinol F (6): Green sticky mass [α]^25^_D_ +178.2 (c 0.001, MeOH); UV-Vis λ_max_ (MeOH) nm (log ε) 208 (5.31), 233 (5.37) and 280 (5.44); 1D and 2D NMR (600 MHz, CD_3_OD) data (see Table 1, Table 2 and Appendix A); HRESI (−) MS *m*/*z* 387.1454 [M − H]^−^ (calcd for C_21_H_23_O_7_^−^; 387.1444).

### 5.4. X-ray Crystallographic Analysis of 2

Crystals were obtained by slow cooled evaporation from MeOH: DCM (1:1) solution; suitable crystals were selected for X-ray crystallographic analysis using an MX1 beamline at the Australian Synchrotron, using silicon double crystal monochromated radiation (λ = 0.71073 Å) at 100 K [25]. The XDS software package [26] was used on site for data integration, processing, and scaling. SADABS [27] was used to apply an empirical absorption correction. Shelxt [28] was applied to solve the structure by the intrinsic phasing method, and a suite of SHELX programs [28,29] were used for refinement, via the Olex2 graphical interface [30]. Crystallographic data of **2** (CCDC number: 2236594) was deposited at the Cambridge Crystallographic Data Centre. Additional crystallographic information is available in the Supporting Information (Appendix A).

### 5.5. Crystal Data for 2

C_20_H_21_O_7_ (M = 373.37 g/mol); triclinic, 0.2 × 0.1 × 0.1 mm^3^, space group *P*1_,_
*V* = 872.6(3) Å^3^, *Z* = 2, *D*_c_ = 1.421 g/cm^3^, *F*(000) = 394.0, Mo Kα radiation, λ = 0.71073 Å, *T* = 100 K, μ = 0.108 mm^−1^; 2θ_range_ = 3.552 to 57.202°, 20,884 reflections collected, 6838 unique (*R*_int_ = 0.0191); final GooF = 1.048, R1 = 0.0383 [*I* > 2σ(*I*)], wR2 = 0.1031; absolute structure parameter = 0.06(13).

### 5.6. Maintenance of RAW 264.7 Macrophages

Cells were grown in 75 cm^2^ flasks on Dulbecco’s Modified Eagle Medium (DMEM) containing 10% fetal bovine serum (FBS) that was supplemented with penicillin (100 U/mL), streptomycin (100 μg/mL) and L-glutamine (2 mM). The cell line was maintained in 5% CO_2_ at 37 °C, with media being replaced every 3–4 days. Once cells had grown to confluence in the culture flask, they were harvested using a rubber policeman, as opposed to using trypsin, which can remove membrane-bound receptors.

### 5.7. Pro-Inflammatory Activation of Cells

RAW 264.7 cells (1 × 10^6^ cells/mL) were seeded in 96 well plates (Corning^®^ Costar^®^, Sigma, Sydney, Australia) overnight until confluency. When the cells were confluent, each compound was serially diluted from a starting concentration of 100 μg mL^−1^ to construct a dose response curve (i.e., 100, 50, 25, 12.5, 6.25, and 3.13 μg mL^−1^) and co-incubated with cells for 1 hr prior to the addition of 1 μg mL^−1^ LPS and 10 U mL^−1^ (1 unit = 0.1 ng/mL) IFN-γ. After activation, the cells were incubated for another 24 hrs at 37 °C. The supernatant was then collected for NO, and TNF-α assays. The cells were subjected to cell viability measurement using the Alamar Blue assay. Non-activated cells (exposed to media only) were used as negative control and activated cells were positive control.

### 5.8. Determination of Nitrite by the Griess Assay

Nitric oxide was determined by the Griess reagent as described in previous studies [31]. Griess reagent was freshly made up of equal volumes of 1% sulfanilamide in 5% phosphoric acid and 0.1% N-1-naphthylethylenediamine dihydrochloride in Milli-Q water. From each well, 50 µL of supernatant was transferred to a fresh 96-well plate and mixed with 50 µL of Griess reagent. The production of nitrite as an indicator of NO was measured at 540 nm in a POLARstar Omega microplate reader (BMG Labtech, Mornington, Australia).

### 5.9. Determination of TNF-α by ELISA

The stored supernatants were analyzed for TNF-α synthesis using a commercial ELISA kit (Peprotech, Brisbane, Australia) according to the manufacturer’s instructions. The absorbance was measured at 410 nm [32]. The concentrations of TNF-α in the experimental samples were extrapolated from a standard curve.

### 5.10. Determination of Cell Viability by the Alamar Blue Assay

After various treatments and the stimulation by LPS and IFN-γ overnight, 100 µL of Alamar Blue solution [10% Alamar Blue (resazurin) in DMEM media] was added to cells and incubated at 37 °C for 2 hrs. The fluorescence intensity was measured with excitation at 530 nm and emission at 590 nm using a microplate reader. The results were expressed as a percentage of the intensity to that of control cells (non-activated cells).

### 5.11. Statistical Analysis

Data analysis was carried using GraphPad Prism 9.3.1. Calculations were performed using MS-Excel version 16.61.1. IC_50_ values were obtained by using the sigmoidal dose–response function in GraphPad Prism. The results were expressed as mean ± standard deviation (SD).

## 6. Conclusions

In conclusion, we have isolated and characterized six new rare peltogynoid flavonoids from the leaves of *Backhousia myrtifolia*. Myrtinols exhibited promising anti-inflammatory activity; however, their low therapeutic index warrants further cytotoxic evaluation as part of a future investigation.

## Figures and Tables

**Figure 1 molecules-28-02160-f001:**
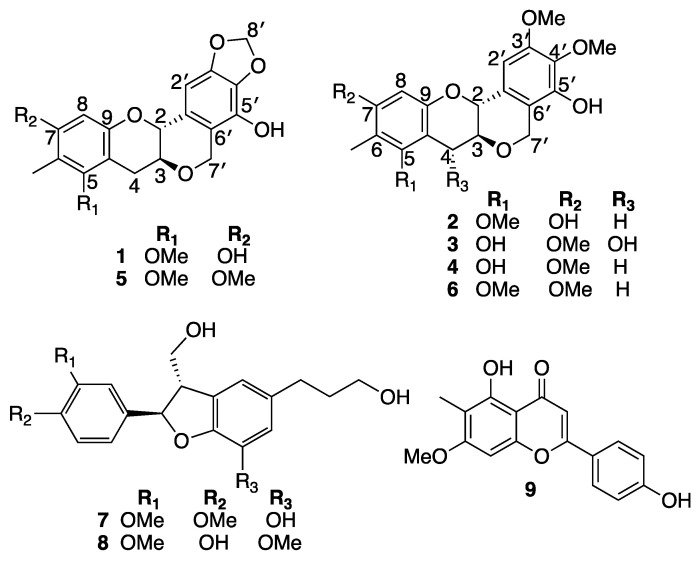
Structures of compounds **1**–**9**.

**Figure 2 molecules-28-02160-f002:**
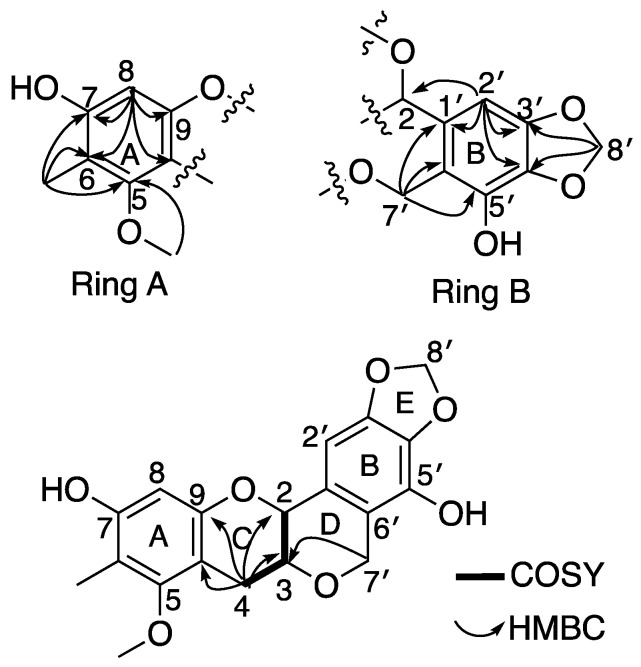
Diagnostic (600 MHz, CD_3_OD) HMBC and COSY correlations of **1**.

**Figure 3 molecules-28-02160-f003:**
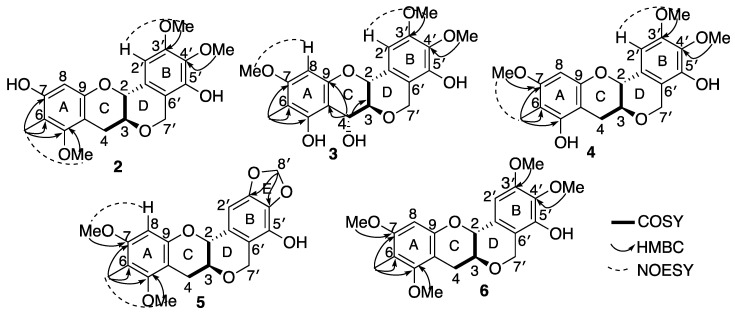
Key diagnostic 2D NMR correlations of compounds **2**–**6**.

**Figure 4 molecules-28-02160-f004:**
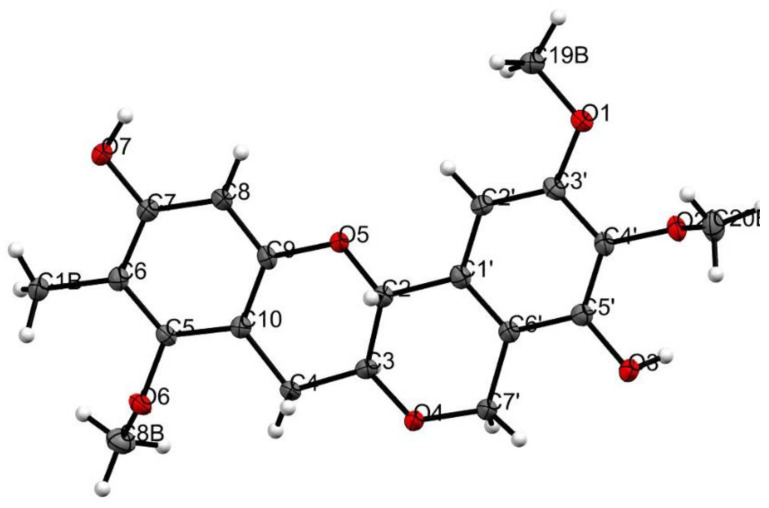
ORTEP diagram of **2**. The compounds in the asymmetric unit are an inversion of each other.

**Table 1 molecules-28-02160-t001:** ^1^H NMR data for compounds **1**–**6**.

Position	δ_H_ (*J* in Hz) 1 ^a^	δ_H_ (*J* in Hz) 2 ^a^	δ_H_ (*J* in Hz) 3 ^a^	δ_H_ (*J* in Hz) 4 ^a^	δ_H_ (*J* in Hz) 5 ^a^	δ_H_ (*J* in Hz) 6 ^b^
2	4.53, d (*9.0*)	4.57, d (*9.5*)	4.74, d (*9.9*)	4.56, d (*9.6*)	4.58, d (*9.1*)	4.67, d (*9.5*)
3	3.64, m	3.67, ddd (*10.7*, *9.5*, *6.1*)	3.67, dd (*9.9*, *8.3*)	3.71, ddd (*10.5*, *9.6*, *6.1*)	3.67, ddd (*10.6*, *9.1*, *5.9*)	3.74, m
4a 4b	3.11, dd (*15.5*, *6.0*) 2.63, dd (*15.3*, *10.7*)	3.15, dd (*15.4*, *6.1*) 2.66, dd (*15.4*, *10.7*)	5.01, d (*8.3*)	3.10, dd (*15.4*, *6.1*) 2.57, dd (*15.4*, *10.5*)	3.14, dd (*15.4*, *5.9*) 2.65, dd (*15.4*, *10.6*)	3.23, dd (*15.3*, *6.0*) 2.71, dd (*15.3*, *10.6*)
8	6.21, s	6.23, s	6.15, s	6.19, s	6.36, s	6.36, s
5-OMe	3.71, s	3.72, s			3.73, s	3.73, s
6-Me	2.05, s	2.07, s	1.98, s	2.02, s	2.06, s	2.09, s
7-OMe			3.77, s	3.76, s	3.80, s	3.80, s
2′	6.66, s	6.78, s	6.79, s	6.83, s	6.72, s	6.84, s
7′a	4.89, d (*15.2*)	4.88, (*15.3*)	4.92, d (*15.1*)	4.89, d (*15.1*)	4.91, d (*15.2*)	4.96, d (*15.4*)
7′b	4.73, d (*15.2*)	4.73, (*15.3*)	4.72, d (*15.1*)	4.74, d (*15.1*)	4.76, d (*15.2*)	4.81, d (*15.4*)
8′	5.92, dd (*15.3*)				5.93, d (*14.0*)	
3′-OMe		3.88, s	3.89, s	3.89, s		3.95, s
4′-OMe		3.79, s	3.81, s	3.79, s		3.91, s

^a^ Recorded in CD_3_OD; ^b^ Recorded in CDCl_3_.

**Table 2 molecules-28-02160-t002:** ^13^C NMR data for compounds **1**–**6**.

Position	* δ_C_ 1 ^a^	* δ_C_ 2 ^a^	* δ_C_ 3 ^a^	* δ_C_ 4 ^a^	* δ_C_ 5 ^a^	* δ_C_ 6 ^b^
2	74.6	74.4	72.7	74.2	74.6	74.3
3	73.1	72.9	78.3	73.3	72.7	72.8
4	28.2	28.1	70.6	28.6	28.0	28.4
5	157.6	158.8	156.8	158.8	158.3	157.4
6	110.5	111.6	106.8	106.5	112.8	112.6
7	155.2	156.2	160.0	158.5	158.8	157.8
8	99.7	99.4	92.1	92.4	101.9	99.6
9	152.9	154.0	154.0	154.2	154.8	153.1
10	105.1	106.8	103.8	102.8	108.0	106.8
5-OMe	60.3	60.2			60.1	61.1
6-Me	8.6	8.4	7.7	8.1	8.7	9.4
7-OMe			60.8	55.8	55.7	56.7
1′	117.7	116.0	116.1	116.3	118.4	114.2
2′	98.4	101.3	101.4	101.0	98.6	102.4
3′	133.4	136.2	135.5	136.2	134.9	134.4
4′	147.6	153.2	153.2	153.3	148.8	147.7
5′	135.7	146.4	147.0	147.1	137.5	
6′	126.9	129.7	129.4	130.1	128.4	128.7
7′	65.8	65.5	65.6	65.5	65.8	65.7
8′	101.9	101.3			101.9	
3′-OMe		56.3	56.2	56.2		57.1
4′-OMe		60.9	60.8	60.9		62.2

^a^ Recorded in CD_3_OD; * assignments supported by HSQC and HMBC; ^b^ Recorded in CDCl_3_.

**Table 3 molecules-28-02160-t003:** In vitro anti-inflammatory activity of compounds **1**–**9** in LPS + IFN-γ activated RAW 264.7 macrophages.

Compounds	Inhibition of NO Production (µg/mL), N = 6	Cell Viability (µg/mL), N = 6	Inhibition of TNF-α Production (µg/mL), N = 6
**1**	11.47 ± 0.14	18.76 ± 0.39	24.54 ± 0.28
**2**	16.25 ± 0.77	37.22 ± 2.09	52.35 ± 7.47
**3**	>100	>100	>100
**4**	29.31 ± 10.95	71.46 ± 14.74	91.26 ± 7.14
**5**	8.51 ± 0.47	15.52 ± 0.42	17.21 ± 0.22
**6**	12.62 ± 0.26	19.79 ± 0.86	30.55 ± 5.01
**7**	44.11 ± 13.39	>100	>100
**8**	>100	>100	>100
**9**	8.30 ± 0.96	20.45	46.79 ± 5.87
**curcumin (+ve control)_**	4.64 ± 1.5	10.86 ± 1.3	4.19 ± 2.6

## Data Availability

NMR data can be obtained by contacting the corresponding author.

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
