# Peer review of "Myrtinols A–F: New Anti-Inflammatory Peltogynoid Flavonoid Derivatives from the Leaves of Australian Indigenous Plant Backhousia myrtifolia"

_molecules, 2023, doi:10.3390/molecules28052160_

Round 1

Reviewer 1 Report

Dear authors,

Thank you for your submitted article, however, there are a few major comments that I feel should be addressed before this article is considered for publication.

1. The introduction is too short and could have more information on peltogynoids and their anti-inflammatory or general activities.

2. 3.7. Pro-inflammatory activation of cells – the authors state that compounds were 2-fold serially diluted from100 µg/mL, to what final concentration?

3. Line 98 – LC50 should be IC50.

4. Table 3 – no significance difference is given 1) between the compounds and the positive control, 2) between the compounds themselves. In text, compounds are stated as better than the other in terms of anti-inflammatory activity, but no statistics given to support this.

5. The anti-inflammatory data (inhibition of NO and inhibition of TNF-α production) is discussed in terms of structure activity relationship but not in terms of comparison to other studies such as those mentioned for example, reference 21 where kinase inhibitory activity is tested.

6. In addition the cytotoxicity of the compounds is not discussed compared to the anti-inflammatory activity as this are done concurrently in the RAW 264.7 cell line. For example, compound 1 has NO inhibition IC50 of 11.47 µg/mL and cytotoxicity of IC50 18.76 µg/mL. Therefore, how much of the NO inhibition is due to the compound and how much is due to cell death (fewer cells available, fewer NO molecules to be produced)? This has to be elucidated for all compounds including the positive control. In addition, no discussion is done compared to other studies, for example, the stated reference 23, where authors studied the cytotoxicity of peltogynoids against various cells.

7. The title of the study is ‘New anti-inflammatory Peltogynoid derivatives from the leaves of Australian Indigenous plant Backhousia myrtifolia’ yet when it comes to the results on anti-inflammatory activity, these results are very understated and not well discussed. This should be improved upon.

8. No conclusion is given, a conclusion should be added.

Reviewer 2 Report

Regarding the manuscript entitled "Myrtinols A – F: New anti-inflammatory Peltogynoid derivatives from the leaves of Australian Indigenous plant Backhousia myrtifolia" I have some comments and suggestions to improve the quality of paper. All are commented in the manuscript file attached.

As general comments: the "Results" section must be sub-divided to make it clearer. The findings must be further discussed, comparing them with similar results previously reported, no studies in this species? There is no conclusion section, it must be added.

Round 2

Reviewer 1 Report

Dear authors,

As you have suggested, please change the title to '“Myrtinols A – F: New Peltogynoid flavonoid derivatives from the leaves of Australian Indigenous plant Backhousia myrtifolia'.

Reviewer 2 Report

The manuscript has been revised according to the recommended/requested items, now can be further considered